# Sediment Characteristics over Asymmetrical Tidal Sand Waves in the Dutch North Sea

**Chiu Hwa Cheng [1,*]** , **Karline Soetaert [1,*] and Bas Wijnand Borsje [2]**

[1]   Department of Estuarine and Delta Systems (EDS), NIOZ Royal Netherlands Institute for Sea Research, and Utrecht University, Korringaweg 7, 4401NT Yerseke, The Netherlands

[2]   Water Engineering and Management, University of Twente, PO Box 217, 7500AE Enschede, The Netherlands; b.w.borsje@utwente.nl

\*   Correspondence: chiu.cheng@nioz.nl (C.H.C.); karline.soetaert@nioz.nl (K.S.)

**Abstract:** The behavior of asymmetrical bedforms, which include many tidal sand waves, is challenging to understand. They are of particular interest since they are mostly located within areas prone to offshore engineering activities. Most experimental investigations regarding asymmetrical bedforms consider the riverine environment, are limited to a single sand wave or a few scattered ones, and focus only on differences between crest and trough. Hardly any information is available on sediment compositional changes along asymmetrical tidal sand waves, despite their abundance offshore. An asymmetrical sand wave field located off the coast of Texel Island in the North Sea was studied in June and October 2017. A total of 102 sediment samples were collected over two seasons along a single transect that covered five complete sand waves to measure the grain size composition, organic carbon concentration, chlorophyll-a (chl-a) concentration, and sediment permeability. We found significant variations in these sediment parameters between the sand wave trough, crest, and gentle and steep slopes, including a difference in permeability of more than 2-fold, as well as a difference in median grain size exceeding 65 μm. Based on these characteristics, a sand wave can be divided into two discrete halves: gentle slope + crest and steep slope + trough. Our results indicate a distinct sediment-sorting process along the Texel sand waves, with a significant difference between the two halves of each sand wave. These data could serve as input for process-based modeling of the link between sediment-sorting processes and seabed morphodynamics, necessary to design offshore engineering projects.

**Keywords:** seabed morphology; permeability; asymmetrical sand waves; North Sea; sediment characteristics; sandy shelf seas; biogeochemistry

---

## 1. Introduction

Tidal sand waves are dynamic rhythmic bedforms, often found in tide-dominated, sandy, shallow coastal regions such as the North Sea, but also in many other environments such as straits and tidal inlets around the world [1–4]. They typically range from 100 to 1000 m in wavelength (distance from crest to crest) and have heights up to 5 m [1,5–7]. However, giant, 10-m high sand waves have also been observed in other locations outside the North Sea [8–12]. One prominent feature of sand waves is their ability to migrate as a result of the residual current or tide asymmetry [13–15]. This movement, which involves up to tens of meters per year, can pose potential hazards to navigation and expose pipelines or buried cables [16]. Offshore engineering activities, such as sand mining, the disposal of dredged material, windfarm construction, shipping, and pipeline and cable installations, are projected to increase in the future. Thus, sand wave mobility is especially problematic for coastal management and calls for a solid understanding of the complete sand wave dynamics [17–20]. A multitude of studies

have utilized models to accurately predict sand wave occurrence and movement [1,2,4,15,18,21–26]. More recently, model studies have focused on the development of asymmetrical sand waves [15,27]. However, as knowledge on the sediment characteristics is limited, these state-of-the-art models do not account for spatial variability in sediment composition (e.g., grain size and roughness) yet.

Bottom roughness and topography are important factors contributing to morphological pattern development [25]. Bedforms such as sand waves emerge because of tidal and wave energy dissipation, which causes instability in the sandy seabed. A balance is established between the steady stream transport from the bottom perturbation interacting with the oscillating tidal current, and the force of gravity dragging sediment downslope. Sand waves consequently grow from the resulting net sediment transport, which converges from troughs to the crests [3,26,28–30]. Sediment composition is directly linked to bottom roughness and thus also affects seabed disturbance rates [23,26,30–33] and sand wave emergence and growth. Furthermore, local hydrodynamic conditions often give rise to differential grain-sorting phenomena that can significantly affect the spatial and temporal development of bottom morphology [34–38] and result in sand waves developing either symmetrical or asymmetrical shapes. The asymmetry (A) is measured by subtracting the distance from the crest to the trough of a gentle slope or steep slope ($L_1$ and $L_2$, respectively), divided by the total length of the sand wave (L); this shows how similar or different the two halves of a given sand wave are (Figure 1). At present there is little information on asymmetrical bedforms due to the difficulties of sampling sediment at the required fine-scale level in the field [39–43]. The few field studies available on sand waves have mostly observed coarser sediments on crests compared to troughs, but did not consistently study the composition of slopes. Also, the focus has been on grain size rather than on related components such as biogeochemical compounds [3,5,21,34,44–49].

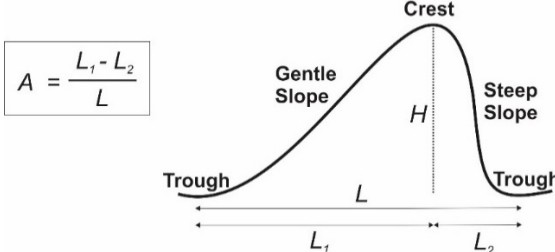

**Figure 1.** Schematic of an asymmetrical sand wave, with the equation for calculating the level of asymmetry (*A*). *H* represents the sand wave height and, similar to the length, is in meters. *A* is a dimensionless value.

In this study, we sampled multiple asymmetrical sand waves in one specific location over two seasons and measured the grain size composition, the organic carbon and chlorophyll-a (chl-a) concentration, and the sediment permeability. Organic carbon and chl-a are important biogeochemical compounds because they support a wide range of metabolic and chemical activities within marine sediments, and are a food source for microbial and benthic organisms [50,51]. While chl-a is a useful measure of the readily consumed (e.g., labile) fraction of organic matter [52–56], organic carbon comprises a mix of fractions, including the more refractory compounds. Sediment permeability is increasingly recognized as an important parameter driving biogeochemical and small-scale water transport processes, especially in the upper layers of sandy sediment and around small bedforms and protrusions [57]. Particularly in sand, it is an important parameter that determines the transport of solutes and fine particles in the sediment [58–60]. Sediment permeability is closely related to the sediment grain size and the associated biogeochemical compounds within the finer fractions [61].

The aim of this study is to capture consistent, small-scale variations in sediment characteristics in a field of asymmetrical sand waves. We test whether sediment parameters differ between different sand waves, between positions within a sand wave, and between the seasons.

## 2. Materials and Methods

### 2.1. Sand Wave Symmetry and Morphological Units

Recent data have shown that sand waves are widely distributed throughout the North Sea and are asymmetrical in shape [43]. A re-analysis of these data [43,62] shows the mean asymmetry in the North Sea to be around 0.3, with asymmetry increasing towards the coast and northwards within the Dutch North Sea (Figure 2D). The level of asymmetry of sand waves is defined as the difference between the length of the gentle slope (gradual, longer side, $L_1$) and the length of the steep slope (steeper, shorter side, $L_2$), divided by the total length of the given sand wave (i.e., 0 is fully symmetrical, 1 is fully asymmetrical; Figure 1). We utilized the same approach for calculating the asymmetry of the Texel sand waves.

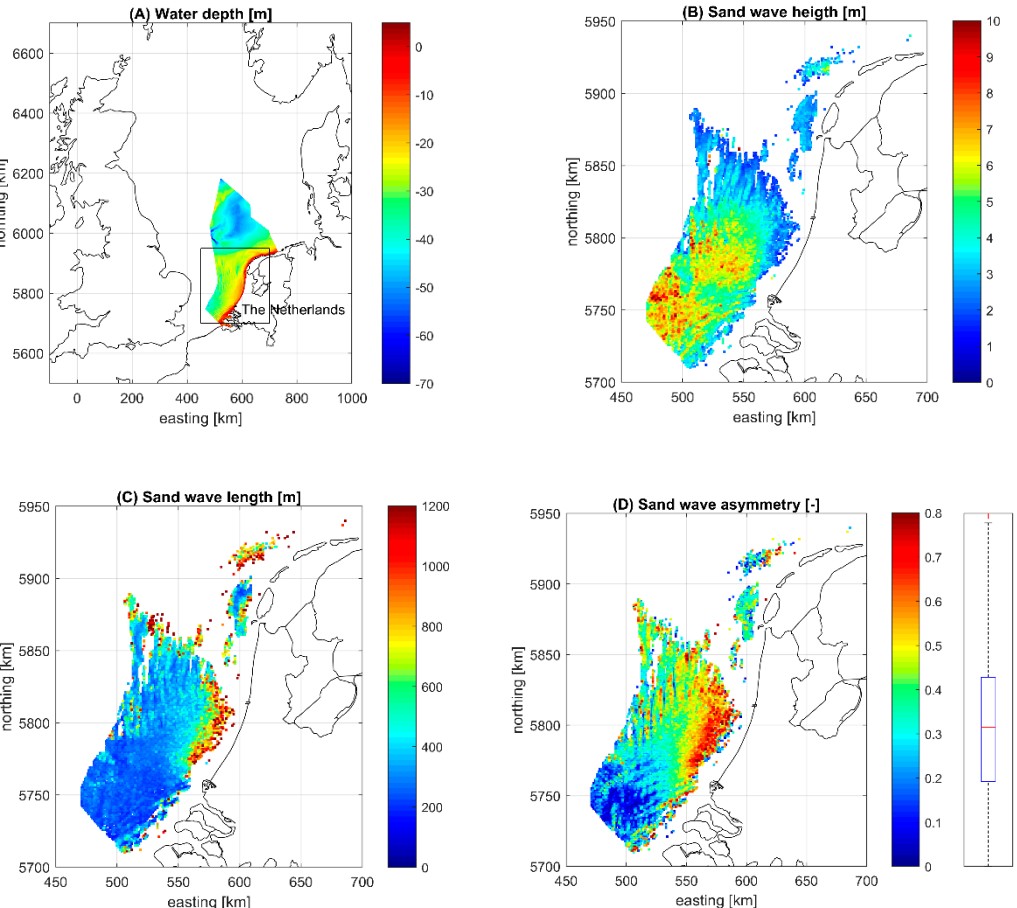

**Figure 2.** (**A**) Map of the North Sea, with the bathymetry information included for the Dutch region Maps showing the height (**B**) and the length (**C**) of the sand waves in the Dutch region. (**D**) Map showing sand wave distribution throughout the Dutch North Sea, as well as the level of sand wave asymmetry. The boxplot on the right shows the distribution of all the measured sand waves by their shape, with extreme examples ranging from 0 to almost 0.8 (outliers excluded). The most asymmetric sand waves are found near to the coast and towards the north. The data were extracted from a study on the environmental properties of sand waves, and aggregated per square kilometer [43,62]

### 2.2. Study Site

Two field campaigns to a sand wave field located in the Dutch North Sea, approximately 22 km offshore of Texel, were undertaken onboard the research vessel, NIOZ *RV Pelagia,* in June and October 2017 (Figure 3A). The particular sampling site was chosen as the mean asymmetry of these sand waves fall within the mean asymmetry value of all sand waves found in the Dutch North Sea (Figure 2D).

Based on the equation from Figure 1, our sand waves had an asymmetry value ranging from 0.29 to 0.38. In addition, as there are shipping lanes located close to the study site, the area is subject to low levels of fishing activity [27]. This greatly reduces the occurrence of activities such as trawling, which can disturb the seabed morphology and other biogeochemical processes [63]. The sand waves extend for another 1-1.5 km to the north and west of our sampling area.

## 2.3. Multibeam Data Collection and Selection of Sampling Locations

The sampling area (~1 km × 3.5 km) was mapped with a Kongsberg EM302 Swath Multibeam echo sounder (MBES) immediately prior to each sampling operation (N 53° 11.241'; E 4° 28.628'). The data from the MBES were processed with the software *Fledermaus* at 1-m grid resolution, with an estimated accuracy of greater than 0.6% of the depth (values supplied by manufacturer). The data contained spikes (noise) and lines that were parallel to the navigation direction due to errors in the speed of sound or from the bottom depths not being corrected for the tidal height. The data were processed with *CUBE* (Combined Uncertainty Bathymetric Estimator), which estimates the most probable position of the bottom. The spikes in the data had to be manually removed. The resulting bathymetric data were converted to a raster using R 3.4.4. [64], with the function *rasterFromXYZ* from the *raster* package (R. J. Hijmans, Davis, United States) [65].

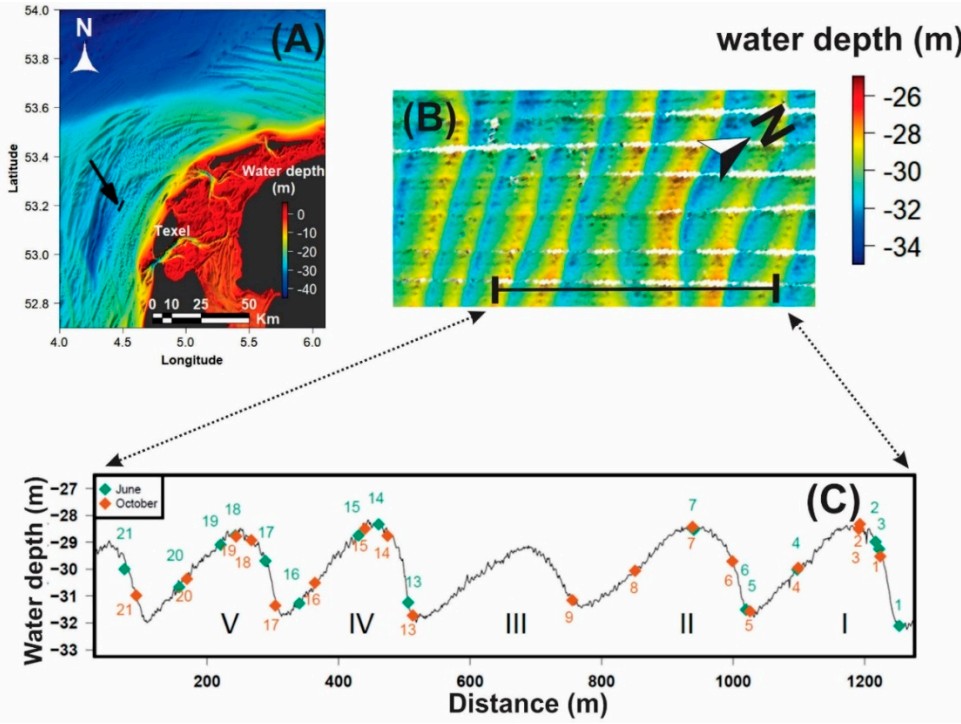

**Figure 3.** (**A**) Map showing the study location, offshore of Texel [66]. (**B**) Subset of the mapped area. (**C**) The profile was created using the processed and gridded geophysical data collected with a MBES from June, and shows a cross section of our sampling transect, with stations from both seasons plotted together.

During the MBES mapping in June, the sailing speed of the vessel was approximately 5 ± 1 knots. The temperature and salinity averaged 14.6 ± 0.8 °C and 35.2 ± 0.02 ppt. Wind speed averaged 1.7 ± 0.7 m s$^{-1}$. In October, the sailing speed was approximately 2.2 ± 1.7 knots, while the temperature and salinity averaged 15.3 ± 1.9 °C and 34.9 ± 0.02 ppt. Wind speed averaged 2.8 ± 0.9 m s$^{-1}$. For additional information about the weather conditions immediately preceding and during the campaigns, including for each type of sampling, please see Figure S1. The *RV Pelagia* does not have a tidal correction

mechanism or a DP (dynamic positioning system), but is equipped with a GNSS (Global Navigation Satellite System) positioning system.

### 2.4. Sediment Sampling

All samples were collected along a single transect line (~1100 m) covering five sand waves (Figure 3C). Given the time limitations of the sampling, we positioned our stations only along the four highest sand waves (I, II, IV, and IV). The middle sand wave (III) was excluded from sampling because it was shorter than the adjacent sand waves, and we wanted to measure sand waves exhibiting the steepest gradients possible. The first campaign included 16 stations (1–7 on sand waves I and II and 13–21 on sand waves IV–V). Sediment was collected using the NIOZ multicorer (Oktopus model), equipped with eight 10 cm diameter cores, each 61 cm in length. The multicorer is neither a piston corer nor a vibracorer, so no additional mechanical forces are involved in the sediment sampling. From each of these cores, subsamples were taken by pushing a single 3.5 cm diameter core into the multicore sediment core, as close to the center as possible. Three subsamples were taken for grain size composition and organic carbon concentration, with another three cores subsampled for permeability. In June, the core lengths ranged from 11 to 19 cm in length. In October, the core lengths ranged from 4 to 17 cm in length. From the remaining multicore, the top 1 cm of sample was collected with a 60 mL cutoff syringe for the chl-a concentration (only one sample per station). Usually, a single deployment of the multicore was sufficient to collect all the subsamples per station. However, stations 19 and 21 were sampled with a 30-cm diameter NIOZ Box corer (K6 model) because of repeated multicorer failures due to sediment coarseness. The dimension of the box corer was approximately 32 cm in diameter and 55 cm in length. All six subsamples were taken directly from a single box core sample from the respective stations. In the second campaign, 18 stations (1–9, 13–21) were sampled along the same transect (extra stations: 8 and 9 on sand wave II), using the same exact sampling protocol. However, the NIOZ multi-corer was only used for stations 1, 2, 4, and 5. The remaining stations were collected with the NIOZ Box corer. Sampling was executed in relatively calm weather, with waves at most 1 to 1.5 m in height in June and somewhat higher in October at above 2 m (Figure S1).

As we expected to find a difference in the sediment parameters along the sand waves, we subdivided them into the following morphological units (MUs): crest, trough, gentle slope, and steep slope. We set the boundary of the two slopes as the regions between 0.5 m above the troughs and 0.5 m below the crests (Figure 3C).

### 2.5. Sediment Samples: Grain Size Distribution, Organic Carbon, and Chl-a

The sediment subcores were collected and kept stable in a cool location until further processing was possible. The sediment samples were stored in plastic vials and frozen at −20 °C, weighed, freeze dried, and dry weighed; porosity was measured on 24 random samples. Samples were analyzed for grain size composition using the Malvern Mastersizer 2000 particle size analyzer (Malvern, (WM), United Kingdom) through laser diffraction [67], which measured the volume percentages of five sediment fractions: coarse sand (500–1000 μm), medium sand (250–500 μm), fine sand (125–250 μm), very fine sand (62.5–125 μm), and silt (≤63 μm). Together, these five fractions total 100% for each sample. The median grain size (D50) was calculated from this information.

Additional sediment samples were analyzed for the total organic carbon concentration. Approximately 50 mg from each sample was added to silver capsules and acidified with several additions of HCl to remove the carbonate content. The samples were then measured on a Flash2000 organic elemental analyzer (Bath, (SW), United Kingdom) [68].

The chl-a sediment samples were immediately stored in a −80 °C freezer upon collection, and later freeze-dried overnight back in the lab under dark conditions at −60 °C. The samples were then extracted with 90% acetone and measured on an Analytik Jena Specord 210 spectrophotometer (Jena, (TH), Germany) with a halogen lamp [69].

The average length of the sediment cores was 13 cm in June and 9.5 cm in October. The results presented here are averages of the entire individual sediment cores.

*2.6. Sediment Permeability*

Sediment permeability was measured using a constant head permeameter [70,71]. The setup measures the hydraulic conductivity of sediment cores, which is used to calculate permeability as a function of the following equation:

$$k = \frac{K \times u}{d \times g},$$ (1)

where $k$ is the permeability (m$^2$), $\mu$ is the water viscosity (Pa*s), calculated from the temperature and salinity, $d$ is the water density (g cm$^{-3}$), $g$ is the gravitational acceleration (9.81 m s$^{-2}$), and $K$ is the sediment hydraulic conductivity (cm s$^{-1}$), calculated as follows:

$$K = \frac{V \times L}{h \times A \times t},$$ (2)

where $V$ is the water volume collected from the core (cm$^3$), $L$ is the sediment length (cm), $h$ is the pressure difference between the reservoir and outlet (pressure head; cm), $A$ is the core cross-sectional area (cm$^2$), and $t$ is the time to collect $V$ (s).

*2.7. Statistical Analysis*

The effects of the MU, sand waves, and seasonality on the permeability, D50, the five sediment fractions, the organic carbon, and chl-a concentrations were tested using a three-way ANOVA (Analysis of variance) test. This test looks for any interactions and effects between three independent variables and a continuous dependent variable.

To determine the cause for any statistically significant differences, a post hoc pairwise comparison (Tukey's HSD honestly significant difference test) was run. The purpose of this is to identify where the significant differences may stem from, by comparing the means of every treatment with every other treatment to identify differences between any two means that would be greater than the expected standard error. The test compared all the MUs with one another over each sediment parameter. All statistical analyses were conducted in R, with the *lsmeans* package (R. V. Lenth, Iowa City, United States) [64,72].

## 3. Results

The Texel sand waves are largely asymmetric (Figure 3C), with the steep slopes oriented NNE (north-northeast), and the studied sand waves ranging from 2.8 to 3.5 m in height and 160 to 210 m in length. Out of 16 stations in June, two were located in the gentle slope, five in the crest, five in the steep slope, and four in the trough. In October, those numbers were four, seven, two, and five, respectively.

*3.1. A Note about the MBES Datasets*

The cross section shown in Figures 3C and 4F was produced using the MBES data from June, as the MBES in October could not generate the bathymetry at equal resolution. The MBES mapping occurred over a three-day period at different times with respect to the tides in October, showing a considerable number of spikes at the interface.

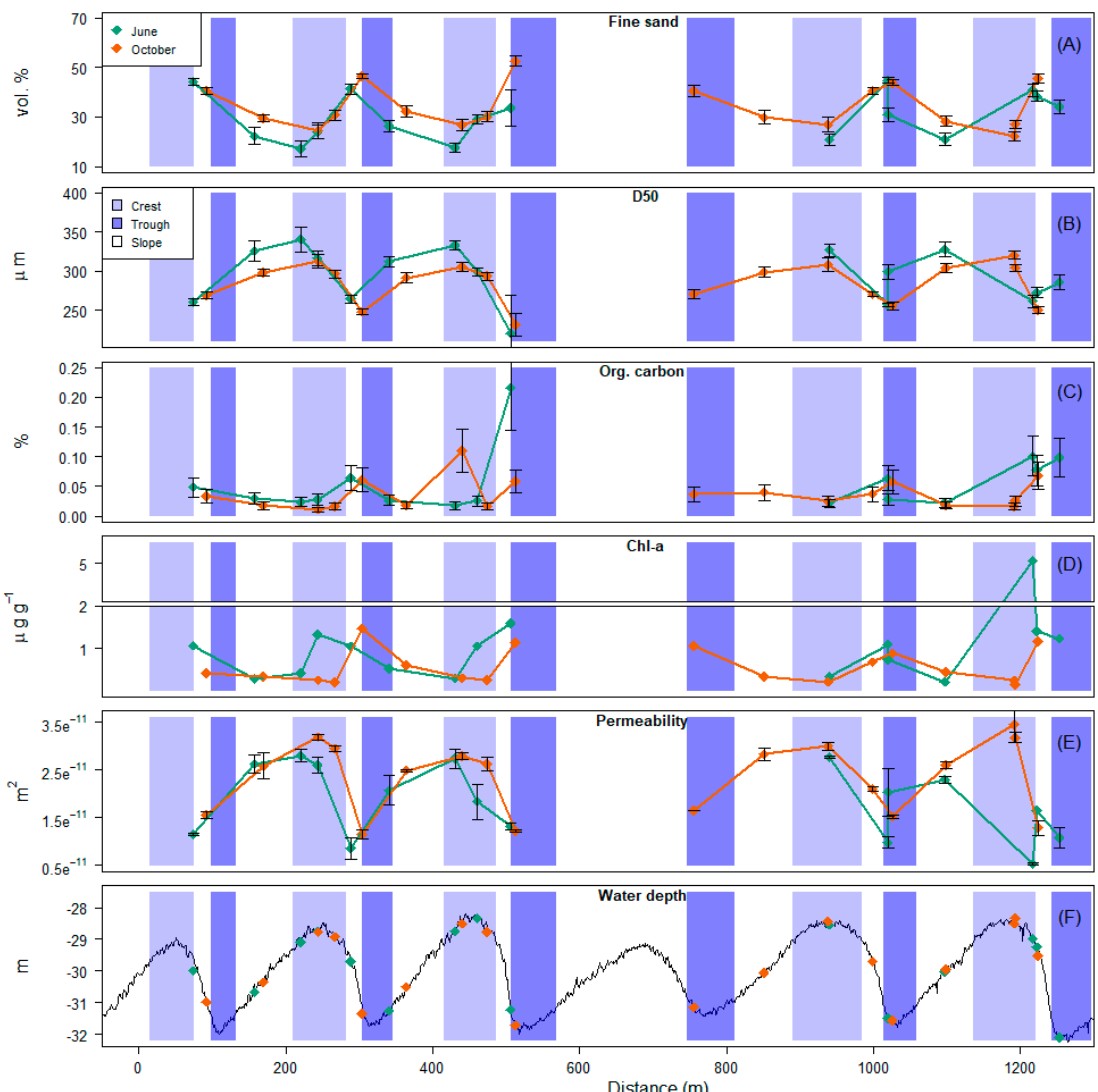

**Figure 4.** Selection of measured sediment parameters along the transect (mean and standard deviation) and for the two seasons. (**A**) Percent fine sand, (**B**) D50size, (**C**) organic carbon concentration, (**D**) chl-a concentration (a break in the figure is used to "rescale" the outlier), (**E**) sediment permeability. (**F**) Transect cross section, with the sampling positions from both campaigns.

### 3.2. Sediment Composition, Organic Carbon, and Chl-a Content

The percentages of fine sand, D50, and sediment permeability all show a clear pattern along the sand waves, which is consistent during both seasons (Figure 4). The sediment composition coarsens from the gentle slope side up towards the crest and decreases again down the steep slope to the trough. The finer fractions (≤250 μm), are more abundant in the steep slope and trough locations and vice versa for the coarser fractions. A similar trend was also observed for the other sediment fractions (Table S1 and Figure S2). The silt fraction was highest in the steep slope and trough stations, as was the very fine sand fraction, but the latter was also found in very small quantities (<0.5%) in the crest and gentle slope areas. The D50 in June averaged 288.9 ± 34.8 μm, with a range of 220–340.5 μm, while in October the average was 284.7 ± 25.8 μm, with a range of 231.1–320.5 μm. The largest difference in average D50 within a sand wave was 66.4 μm in June and 57.2 μm in October. The average D50 was higher in June except for the gentle slope (Table S1). The largest fraction of sediment was fine sand in the steep slope and trough, while in the crest and gentle slope it was the medium sand fraction (Figure 4). Higher-resolution information on the grain size distribution is available as sediment cumulative distribution plots (Figure S3).

The organic carbon concentration values were low overall, with the highest average falling below 0.25%. Nevertheless, the organic carbon showed variations along the sand waves, following the trend of the finer fractions of sediment (≤250 μm). The chl-a content also showed a general pattern of increase up the gentle slope towards the crest, but generally had a peak in the steep slopes, followed by the troughs with almost an 8-fold difference between the steep and gentle slope averages.

Permeability was highest on the gentle slopes in June, while in October the most permeable MU was the crests (Table S1). The permeability values were somewhat higher in October, with a range of $9.49 \times 10^{-12}$–$4.28 \times 10^{-11}$ m$^2$ and an average of $2.35 \times 10^{-11}$ m$^2$. In June, the range was $4.35 \times 10^{-12}$–$3.05 \times 10^{-11}$ m$^2$ and the average was $1.88 \times 10^{-11}$ m$^2$. However, all the measured samples fell within the range of the threshold that is considered permeable ($\geq 10^{-12}$). Overall, the permeability in the crests was more than double that of the troughs, and was closely correlated to the D50. Sediment porosity averaged $0.37 \pm 0.048$ in June and $0.36 \pm 0.067$ in October, with no significant differences between the seasons and no measurable differences along the sand wave.

### 3.3. Trends between the MUs

Averaged over the MUs, the grain size composition and level of permeability were unevenly distributed along the sand waves, with the finest fractions completely absent from the gentle slope and crest (Figure 5). The three-way ANOVA showed that in none of the cases did the individual sand waves themselves have a significant effect on any of the measured sediment parameters (Table S2). The four MUs, when compared between each sand wave, showed no statistical difference to their counterparts ($p > 0.05$). The crest from one sand wave was not significantly different from the crests from other sand waves, and similar for the other MUs. However, all four of the MUs were significantly different from one another ($p < 0.01$) for every measured parameter, while seasonality was highly significant for permeability ($p < 0.01$), the D50, and the coarse and fine sand fractions. For the other parameters, the seasonality was not significant (Table S2). The statistical difference was larger ($p < 0.01$) when the sand waves were split into two halves (gentle slope/crest and steep slope/trough), compared to the tests that considered four individual MUs. Additional information can be found in Figure S4.

The post hoc pairwise comparison (Tukey's HSD honestly significant difference test) showed that in all cases the steep slope was not as significantly different from the trough and, likewise, the crest and gentle slope MUs were less significantly different from each other. In comparison, all of the other combinations were more significantly different. In accordance with the ANOVA results, this means that the crest and gentle slope, while still statistically significantly different from each other, are much more different to the steep slope and trough. Similarly, the trough and steep slope are less significantly different from one another than they are to the crest and the gentle slope.

### 4. Discussion

Given the scarcity of field information about sediment characteristics across asymmetrical sand waves, the aim of our field campaigns was to determine the sedimentological properties of these bedforms. Therefore, we discriminated between different MUs (gentle slope, crest, steep slope, trough) and showed that this leads to measurable changes in the sediment characteristics. On the one hand, our results are consistent with previous field studies in that the sediment is mostly coarser in crests than in the troughs [34,44–47]. However, in addition to these studies [3,5,48,73], we improved upon the sampling resolution by dividing the sand waves into smaller sections (e.g., 4 MUs). Whereas it was difficult to equally sample all MUs from every sand wave (Figure 3C), we successfully sampled many stations (16 in June; 18 in October) along four sand waves, and collected 102 samples in total. Bathymetric data were recorded during both campaigns, but we were not able to determine the actual migration rate of our sand waves due to the vessel's technical limitations (see Section 2.3) and also the many known sources of possible error, such as positioning, bed level noise, and depth distortions. However, previous calculations on several sand wave locations with similar orientations within the Dutch North Sea show a migration rate ranging from less than 1.0 to 8.4 m yr$^{-1}$, with an error of up

to 2 m [74], and some of these figures may even have been overestimated [75]. Although information about the migration rate of the Texel sand waves is not presently available, it is a relevant aspect that should be taken into consideration by future studies concerning dynamic bedforms.

The difference in sediment median grain size ($D_{50}$ in µm) between crests and troughs was up to 25% of the mean and much greater than found in sand waves of comparable size and dimensions from other studies [3,5,48]. Moreover, the sediment coarsened from the gentle slope towards the crest, while the three finest fractions of sediment exhibited significantly higher percentages on the steep slope and trough sections (Figure 5, right panels). These results suggest that sediment sorting along the asymmetrical bedforms effectively creates a distinct gradient between the crest and trough boundary, subdividing the sand waves into two halves: the gentle slope up to the crest, and the steep slope down to the trough. Our higher-resolution sampling thus clearly demonstrates that asymmetrical sand waves are highly complex bedforms with respect to sediment granulometry, permeability, and biogeochemistry.

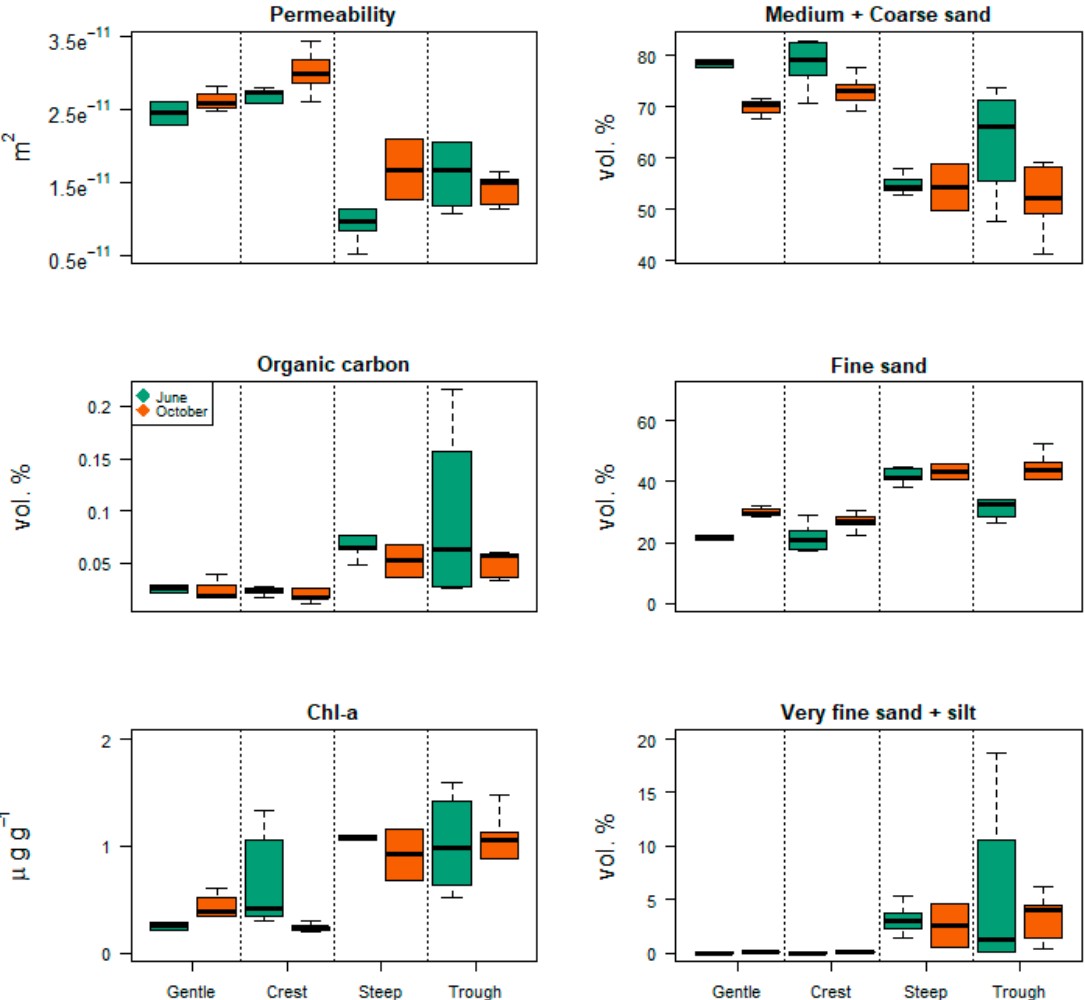

**Figure 5.** Comparison of selected sediment grain size parameters, organic carbon, chl-a, and permeability between June and October, averaged over the sand wave MUs. The coarse-medium fractions and the very fine sand-silt fractions were combined as these pairs followed the same trends over the MUs.

Sediment permeability generally correlates positively with increasing grain size [33]. In line with this, permeability was about twice as high on gentle slopes compared to the troughs in the summer, while in autumn the largest difference was between the crests and troughs. Permeability is a measure for the degree to which currents and waves penetrate the sediment; this has a crucial effect on the sediment

nutrient or oxygen levels and the organic carbon concentrations. It has been shown that in sediments with a permeability of $\geq 10^{-12}$ m$^2$, advective transport mechanisms become significant in regulating biogeochemical processes [57,61,76]. While all of our samples exceeded that threshold, the greater than two-fold difference means that solute transport could potentially occur at twice the rate in the coarsest areas of the sand wave. This also suggests that subtle biogeochemical differences might be found between the more permeable gentle slope and crest part of the sand waves, compared to the steep slope and trough.

Organic compounds are known to accumulate more in finer sediments; in particular, their concentrations are related to the mud and silt fractions [54,77–79]. In line with this, we found the chl-a and organic carbon concentrations to be significantly higher on the steep slope and troughs. Chl-a is a measure for the readily consumed (e.g., labile) fraction of organic matter, so higher concentrations can support higher overall metabolic activity in the sediment. This is consistent with the higher benthic biomass and activity found in the trough [27]. Furthermore, it is also relevant here, since benthic organisms can redistribute the upper layers of sediment, enabling the burial of fine material. A video transect study on the crest-trough comparisons showed four times more animals living on the sediment (epibenthos) and 30 times more holes (a proxy used for endobenthos) in the troughs, although the species composition could not be quantitatively determined based on this information [27]. In addition, [27] observed either highly irregular bedforms, or sometimes the complete absence of them in the troughs of the Texel sand waves. Such smaller bedforms are often superimposed on sand waves and are important as they alter the bed roughness and can slowly migrate along the gentle slope [4,9,80].

Differences among MUs were clearly observed over all sand waves and during both seasons of sampling. Surprisingly, however, there were also some notable differences between June and October, although they were less pronounced than the spatial gradients. There was significant seasonal variability in some sediment fractions, sediments being coarser in June, while both the organic carbon and chl-a content were significantly higher in June. The seasonal differences observed in sediment composition could be due to extreme weather conditions (e.g., storms), which may have winnowed the sediments preceding our June campaign, while fine particles may have accumulated in the sand between the June and October field campaigns [5,81]. The higher chl-a concentration might be related to the spring phytoplankton bloom being deposited on the sediment preceding our June campaign [53,82,83]. Contrary to expectations, however, sediment permeability was found to be lower in June, notwithstanding the sediments being coarser. As a consequence, the relationship between permeability and D50 is ambiguous, showing a dependency on both the grain size and the season (Figure 6). This seems consistent with studies pointing to factors other than D50 that can also affect permeability, such as the grain shape, type of sediment, or sediment structure [84–86]. It is also possible that the seasonal dependency is linked to the higher concentration of organic carbon in June, as organic compounds in the silty fraction can alter the cohesiveness of the sediment [54,77–79,87]. Biogenic substances such as chl-a, organic matter, and biofilm compounds (e.g., extracellular polymeric substances) can increase sediment flocculation or cohesion at the microscale level and have consequences on transport processes on local or even larger scales [51,87–89]. Although this was beyond the scope of our study, other properties such as sediment structure and type should also be considered in future studies, as they could have an effect on grain size determinations [86,90,91].

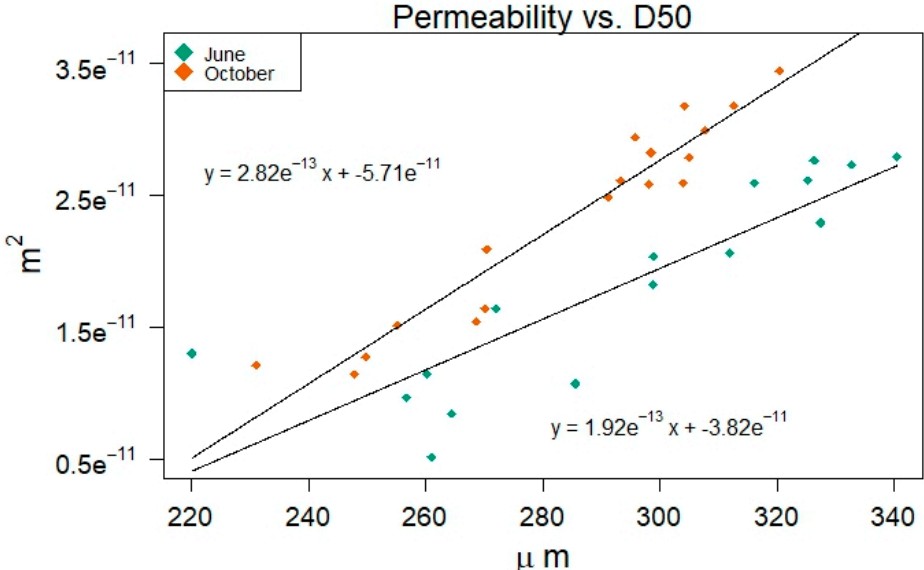

**Figure 6.** The relationship between permeability and the D50 in the two seasons. The individual regressions are shown.

Tidal sand waves are ubiquitous bedforms and occur over a broad range of dimensions in sandy shelf seas. Our field results demonstrate that asymmetrical sand waves exhibit significant spatiotemporal complexities in sediment sorting over small spatial scales, due in part to their irregular shape. While we found consistent spatial trends over two seasons, there is also evidence that these trends are modulated by other factors, probably of biological or biogeochemical origin.

## 5. Conclusions

The sorting of sediment along sand waves with an asymmetry value of 0.29–0.38 was studied at a sand wave field in the Dutch Continental Shelf over two seasons. By classifying the sand waves into four morphological units (gentle slope, crest, steep slope, and trough) we were able to demonstrate significant differences in the sediment grain size, organic carbon concentration, chl-a content, and permeability. The average D50 differed by more than 20% (>60 μm difference in June and >50 μm in October), and permeability by more than two-fold between the crests and troughs, as well as between the crests and steep slopes. This was even more pronounced for the biogeochemical compounds of organic carbon and chl-a, with differences from 4- to 8-fold. Moreover, all of these patterns were observed over two seasonal campaigns. This study sheds light on the small-scale processes that couple the dynamics of sand wave morphology and sediment characteristics, and contributes information previously unavailable to help improve physical or ecosystem models to better understand these environments.

**Supplementary Materials:** The following are available online at http://www.mdpi.com/2077-1312/8/6/409/s1. Supplementary Figure S1: Wave height and mean wind speed during each campaign. Supplementary Figure S2: Additional grain size information. Supplementary Figure S3: Sediment cumulative distribution plots. Supplementary Figure S4: Other grain size parameters. Supplementary Table S1: Sediment results. Supplementary Table S2: Statistical results.

**Author Contributions:** The listed authors in this manuscript have each contributed significantly to the collection of data, analysis and interpretation of the results, and drafting of the paper. Furthermore, each author has contributed substantial time to the editing and revising of each individual section within the manuscript. The study was initiated by the first and third authors, C.H.C. and K.S., who were also fully involved in the methodology, field campaigns, and sample collection. Data analysis was primarily conducted by C.H.C., but K.S. and B.W.B. contributed as well. Discussions of the manuscript content involved all of the co-authors (C.H.C., B.W.B. and K.S.) throughout the drafting and interpretation of the contents. Each co-author also contributed to the final stages of the writing and review. All authors have read and agreed to the published version of the manuscript.

**Funding:** This research was funded by the SANDBOX project, which is a part of NWO-ALW (Nederlandse Organisatie voor Wetenschappelijk Onderzoek-Aard-en Levenswetenschappen). The Royal Boskalis Westminster N.V., the Royal Netherlands Institute for Sea Research (NIOZ), and Utrecht University are acknowledged for their financial support of this project.

**Acknowledgments:** Many thanks to Erik Hendriks, Johan Damveld, Sarah O'Flynn, Justin Tiano, Karin van der Reijden, Leo Koop, Rob Witbaard, Pieter van Rijswijk, and Anton Tramper, who contributed to the sample collection and data contribution for this project from the cruise campaigns. We are very grateful to Texel NMF staff for their support in making both research campaigns successful, and to Henk de Haas for his help with the MBES data and other technical details. Thanks, also, to Peter van Bruegel for the analysis of the sediment samples, Natalie Steiner for helping with the statistical analyses, Jaco de Smit for the cumulative grain distribution analyses, and the many other colleagues who offered assistance along the way. The data collected for this publication can be found in the 4TU Centre for Research Data repository at the following link: 10.4121/uuid:9f6e21c5-f35b-4bca-a468-7534e04bb240. Lastly, we want to thank the reviewers for all of their input, which aided us in further improving this manuscript.

**Conflicts of Interest:** The authors declare no conflict of interest. The funders had no role in the design of the study; in the collection, analyses, or interpretation of data; in the writing of the manuscript, or in the decision to publish the results.

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
