# Peer review of "Sediment Characteristics over Asymmetrical Tidal Sand Waves in the Dutch North Sea"

_jmse, doi:10.3390/jmse8060409_

Round 1

Reviewer 1 Report

The manuscript by Cheng et al. “Sediment characteristics over asymmetrical tidal sand waves” provides new data on asymmetrical sand waves close to the Dutch coast in the North Sea. This is interesting, new, and relevant data on large bedforms that are not easy to sample. The authors show how granulometric and geochemical parameters show clear differences in the different discrete units of the sand waves.

The paper is an easy read in good English, well structured with most of the necessary literature and suitable figures. The discussion could be a bit more substantial/broader.

In some places the ms. can be shortened, in other places some more information is necessary. After revision of some unclear points this paper should be in a publishable form.

Here are some points:

  • Not enough information on the circumstances of the surveys (weather, sea state, tides, accuracy of positioning/sampling, multibeam-survey speed, etc ..)
  • Too much information on some methods, may be reduced
  • No multibeam data in the results section
  • Too much statistics with too few data (?)
  • Not enough advantage taken out of grain-size data

There are more points in the following text that usually start with the line number:

3 Maybe you should add “… in the Dutch North Sea” to the title?

Introduction section: I wonder if everybody has a clear imagination as to what an asymmetrical sand wave actually is. Maybe there should be a short and very clear definition of an asymmetrical sand wave, probably along with a schematic figure to explain asymmetry parameters and units. There is more explanation in Section 2.1 but maybe this information should occur earlier in the manuscript?

79 change to “ … field of asymmetric sand waves.”

81-84 unimportant information (standard structure of a paper), should be deleted.

87-94 Figure 1 and caption: What means “inset of the Dutch region”? B is just a frame that is zoomed up in C (nearly at least, B should laterally end at 700 km Easting). I find C hard to understand. Here should be at least also the straight bathymetry of the area first. It is in Fig. 2, maybe the upper panel of Fig. 2 should come with Fig. 1 or you completely combine both figures? What means the [-] after the title of Fig. 1C?

116 Figure 2. Include axis labels to the upper left figure, add north arrow, mark Texel, and improve the legend. Both upper figures would need a scale bar. Upper right figure legend needs a title. Should there be a reference for the upper left figure data?

117-119 How come the cross section is “interpolated“?

123-124 …”mean asymmetry in the North Sea”? What does that mean?

129 “greater than” or “better than”? Better information is needed: tidal correction (how?), GPS accuracy (RTK?). Accuracy vertical lateral? It seems important to know about these parameters as this is a high-resolution study. What exactly means processed at 1-m grid? The data can be processed in any resolution, but what is the data resolution. In this context survey speed and also sea state during the surveys should be provided.

132 and Fig.2 Why don’t you show the same transect as measured on the second survey? You should at least address why you don’t in the text to not leave the reader to speculate about that. To address that in the discussion is too far away.

134 It is not quite clear why you left out Wave III. It is also asymmetric and it would be interesting to see what happens on this one. So if you have data from this wave as well (6 stations in June, 4 stations in October, as it seems)  I would strongly recommend to include them or to better explain why you decide to leave them out.

135 It is necessary to write a few lines about the conditions during the campaign (see state, tide, weather, wind, waves). Weather information of the days prior to the campaigns and possible storm events in the month prior to the campaigns.

138 winch

144 how long were the subsample cores?

132-159 There is way too much unnecessary description of the sampling gear. However, there is not enough description of the positioning accuracy. How did you make sure that your positioning was correct (gear-on-the-ground position) and how did you know that your sampler was exactly at the wanted position on the profile? How did you manage to precisely figure the sampling stations, given that the MBES were likely processed underway (tidal correction and so on)? A little more on this rather than the weight of a multicorer would be helpful.

170 You don’t measure the mean grain size, you measure the grain size distribution and then calculate the mean.

169-172 Was there no chemical pre-treatment (e.g. remove carbonates?). The Mastersizer almost certainly measured more than these 5 fractions. It is in fact the great strength of the laser particle sizers to measure in very high resolution. You may want to use more than the given fractions for further insight.

173-185 too much detail: can that be cited?

172 You should include here that all grain-size percentage data given in the paper refer to volume percentages.

179 Make new paragraph for the chl-a method.

186-187 Move the core-length information where it belongs (after the sentence that ends in line 144), but leave the ‘average’ information here.

187 You may want to add some information on these cores. Were they more or less homogenous or was there any structure visible? By the way: Is there further “ground-truth” information such as underwater images that would provide information on secondary bedforms?

215 What about the multibeam data? I strongly recommend to show the multibeam data of the two campaigns!

217 better use … “in height” and … “in length” ?

220 Fig. 3: The inset in B should read “Slope” rather than “Gentle/Steep” (also in the Supplementary Fig. 1.

222 Fig. 2 (A) the y-axis label should read “vol. %” (same in the related supplement figure).

226 ff It is clear that related variables show related patterns (e.g. grain-size percentages). Also permeability is related to grain size. Further fractions between fine and medium sand could be interesting, also the grain-size modes and possibly the mean grain size. What about statistic parameters such as sorting and skewness and/or ratios of these and other parameters. It would also be interesting to see the grain-size frequency distribution curves of the different units. After all, you want to gain information on the properties and dynamics of the Mus but you seem to not take advantage of the most relevant granulometric parameters.

240 I wouldn’t call this “significant”.

250 You may add here that the permeability and D50 trends are largely parallel.

274 I would recommend to move the panels vertically closer together and put hairlines between the MUs to make the figure easier to read.

I wonder if mean, median, mode, sorting, skewness and more (see comments above) would yield more or at least substantial further information?

276 … of selected …

281 Most interestingly, there appears to be more silt than fine sand (?) suggesting a bi- or polymodal distribution. Again, I would recommend to show grain-size frequency curves of the different MUs and to work out the differences between them and show them here.

282-299 While I support the use of statistical methods to cast light on large data sets it seems that you are using a sledgehammer to crack a nut. Most of the outcome of the statistics can also be recognized without the statistics. While these findings could be confirmed using statistical methods the universal set (the “n”) is likely not high enough for that (too many variables, too few observations, not evenly distributed).  To increase the number of subsamples taken from one sample from one position on one wave serves other statistics. Here you would need more samples from more sediment waves to get a better universal set. I am not that much into such statistics, hence, I recommend to overthink the statistics and maybe employ statistical tests to confirm that the universal set is sufficient to allow such interpretations.

302 you mean “across” the sand waves, don’t you?

310 Only 4 sand waves because you omitted Wave III.

311-314 This information must occur in the methods section! You should decide to show the transect cross sections from both campaigns (like in Fig. 2) in the results section. If there were technical problems you need to address that. Don’t leave it to the reader to speculate about that.

313 What are the “vessel’s technical limitations” (-> methods section)?

318 Proof that with your multibeam data or explain why you can’t. Why do you estimate exactly 2.8 m ± 2 m?

322 not the fraction but the percentage was higher.

323 … that’s why I wonder why you did not employ the common grain-size statistics like “Sorting”.

331 replace “impacts” with “effect”.

334-338 Interesting point. I recommend to also discuss that in the light of currents (bi-directional tidal), pressure differences, matter fluxes, other … Compare/discuss this area with other areas with similar grain size distribution but without sand waves …

339 … unless they come in flocs in the hydraulic sand fraction and cause unexpected things (papers by Malarkey et al. 2015, Neumann et al. 2019, Parsons et al. 2016, among others).

343 and 344 Two sentences in a row start with “This is …”

346 All authors and all reviewers prefer this citation style: “Damveld et al. 2018” but MDPI doesn’t.

349 “in” the troughs …

348-351 This is indeed an important point! Is there any information on smaller bedforms? Surface roughness can be different as a result of sea state, weather conditions, tide level. Is there any sidescan sonar information (possibly recorded by the multibeam) or even underwater photos or footage taken during the surveys? You should invest a couple of lines to address/discuss roughness issues.

359-361 Weather/sea state in the days/weeks prior to the surveys must be checked to ensure comparable survey conditions. This recommendation came several times. Maybe you can gather the information in the form of a table?

362 here we go! You should check the weather information to make sure!

368 There are also papers on this (e.g. Neumann et al 2016 and references therein).

Reviewer 2 Report

I just have a couple of minor suggestions. They are highlighted in attached file.

Round 2

Reviewer 1 Report

The manuscript benefited from the technical part of the revision process and it would have even more benefited from using more grain-size statistics as stated in the last review. However, the manuscript is now in a publishable state.